# PARK7/DJ-1 as a Therapeutic Target in Gut-Brain Axis Diseases

**DOI:** 10.3390/ijms23126626

**Published:** 2022-06-14

**Authors:** Domonkos Pap, Apor Veres-Székely, Beáta Szebeni, Ádám Vannay

**Affiliations:** 11st Department of Pediatrics, Semmelweis University, 1083 Budapest, Hungary; pap.domonkos@med.semmelweis-univ.hu (D.P.); veres-szekely.apor@med.semmelweis-univ.hu (A.V.-S.); szebeni.beata@med.semmelweis-univ.hu (B.S.); 2ELKH-SE Pediatrics and Nephrology Research Group, 1052 Budapest, Hungary

**Keywords:** PARK7/DJ-1, gut-brain axis, inflammatory bowel diseases, Crohn’s disease, ulcerative colitis, neurodegenerative disorders, Parkinson’s disease, Alzheimer’s disease, blood-brain barrier, oxidative stress

## Abstract

It is increasingly known that Parkinson’s (PD) and Alzheimer’s (AD) diseases occur more frequently in patients with inflammatory gastrointestinal diseases including inflammatory bowel (IBD) or celiac disease, indicating a pathological link between them. Although epidemiological observations suggest the existence of the gut-brain axis (GBA) involving systemic inflammatory and neural pathways, little is known about the exact molecular mechanisms. Parkinson’s disease 7 (PARK7/DJ-1) is a multifunctional protein whose protective role has been widely demonstrated in neurodegenerative diseases, including PD, AD, or ischemic stroke. Recent studies also revealed the importance of PARK7/DJ-1 in the maintenance of the gut microbiome and also in the regulation of intestinal inflammation. All these findings suggest that PARK7/DJ-1 may be a link and also a potential therapeutic target in gut and brain diseases. In this review, therefore, we discuss our current knowledge about PARK7/DJ-1 in the context of GBA diseases.

## 1. Introduction 

There is a growing awareness within the medical and scientific communities about the frequent co-occurrence of gastrointestinal and neurodegenerative disorders.

Indeed, over the past few years, numerous epidemiological and experimental studies have demonstrated that inflammatory bowel diseases (IBD) or celiac disease (CeD) increase the risk of neurodegenerative disorders, including Parkinson’s (PD) or Alzheimer’s (AD) diseases [1]. This pathologic crosstalk between the two organs was described as the "gut-brain axis” (GBA) and is suggested to be regulated through systemic inflammatory and neuronal pathways [2]. IBD, the chronic inflammation of the small and/or large intestine, is characterized by inappropriate activation of the immune response against environmental factors and gastrointestinal dysbiosis resulting in local and systemic inflammation in genetically susceptible individuals [3]. As a result of the intestinal inflammation, plenty of intestine-derived inflammatory mediators, including cytokines and bacterial products, spread via the circulation [2]. It has been suggested that some of these factors can disrupt the blood-brain barrier (BBB), which is a selectively permeable border of the brain microvascular endothelial cells that protect the central nervous system (CNS) against the circulating toxins and pathogens [4]. The impaired BBB then allows the passage of intestine-derived factors to enter the brain thus inducing inflammation and neurodegenerative changes. This is what we call the systemic inflammatory pathway of GBA.

Besides the systemic pathway, however, the existence of a neural pathway is also suggested. Recent experimental results demonstrated that intestinal inflammation leads to increased local alpha-synuclein (α-syn) expression that can be transported retrogradely through the nervus vagus from the gut to the brain thereby facilitating the onset of PD [5]. However, our knowledge about the role of nervus vagus in GBA crosstalk is sparse.

In the past decade, the role of Parkinson’s disease 7 (PARK7/DJ-1) has been demonstrated in neurodegenerative diseases [6]. Indeed, the mutation altering the amount or function of PARK7/DJ-1 leads to the rare, autosomal recessive juvenile form of PD [7]. It has been shown that neuronal cells are more vulnerable to oxidative stress in the absence of PARK7/DJ-1 and the decreased expression of PARK7/DJ-1 leads to a toxic accumulation of misfolded proteins, including α-syn leading to neuronal apoptosis [6]. In addition, pharmacological activation of PARK7/DJ-1 was neuroprotective in animal models of PD, AD, and ischemic stroke, as well [8,9].

Besides the beneficial functions of PARK7/DJ-1 in the CNS, it has recently come into the focus of interest also in connection with gastrointestinal disorders. Our and other studies have demonstrated that intestinal expression of PARK7/DJ-1 is altered in the mucosa of patients with celiac disease or IBD [10]. Furthermore, experimental results revealed that PARK7/DJ-1 affects intestinal inflammation and protects the integrity of the mucosal epithelial barrier as well [11,12,13]. In a recent study, Singh et al. demonstrated that genetic deletion of PARK7/DJ-1 leads to dysbiosis and increased expression of pro-inflammatory molecules in the intestine, and also that of PD-related genes in the midbrain of mice [14]. 

Based on the increasing number of data demonstrating the protective role of PARK7/DJ-1 in both the gut and brain diseases, our review aims to point out how PARK7/DJ-1 may connect the pathologies of the two organs.

## 2. Gut-Brain Axis 

### 2.1. Epidemiological Evidence of a Gut-Brain Axis

IBD, including Crohn’s disease (CD) and ulcerative colitis (UC), affects more than 6.8 million people worldwide with a constantly growing prevalence [15]. It is well known that besides the gastrointestinal manifestations of IBD it can also affect the musculoskeletal system and the skin [16]. More recent epidemiological data also suggest a pathological crosstalk between intestinal inflammation and the development of neurodegenerative disorders, including PD and AD. Indeed, Lin et al. were the first who demonstrated the associations between the development of PD and IBD in a retrospective cohort study [17]. They found that IBD was associated with a 35% increased risk of PD. Within a few years, numerous independent large cohort studies confirmed their original observation. A Danish study demonstrated that IBD patients have a 22% increased risk of PD as compared with non-IBD individuals [18]. Similarly, a Swedish and an American study also found an increased PD hazard ratio in IBD patients compared to controls [19,20]. Also, the meta-analysis of Zhu et al. showed that CD and UC increase the risk of PD by nearly 30% compared to controls [21]. In accordance with the above studies, a South Korean cohort study showed that IBD patients were at a 1.87 times higher risk for PD than controls, respectively [22]. 

Similar to PD, epidemiological studies also revealed an increased risk of AD and dementia in patients with IBD. A meta-analysis by Fu et al. demonstrated that the risk of AD was increased by 52% in patients with gastrointestinal pathologies [23]. In addition, a recently published population-based cohort study demonstrated that the overall incidence of dementia among patients with IBD was significantly elevated (5.5% vs. 1.4% among controls) [24].

Although it is still not clear how intestinal diseases affect the development of CNS diseases, some previous epidemiological observations suggest that the systemic spreading of intestinal inflammation may be involved in the pathological crosstalk between the two organs. Indeed, a study of Peter et al. demonstrated that anti-tumor necrosis factor (TNF-α) therapy of the patient with IBD reduced the incidence of PD by 78%, as well [19]. Moreover, a South Korean study found that in IBD patients receiving anti-TNF therapy did not develop PD, and corticosteroid therapy also reduced the risk of PD by 92% among CD patients [22]. Moreover, a lower risk of PD was demonstrated in IBD patients treated with anti-inflammatory mesalazine or its derivative sulfasalazine [25]. 

Besides IBD, the role of CeD and gluten sensitivity has been suggested in the development of neurological and psychiatric disorders, including epilepsy, anxiety, depression, autism, and schizophrenia, as well [26]. Although our knowledge is sparse it is easy to accept that, similarly to IBD, the systemic spreading of inflammation may contribute to the development of CNS diseases. 

All of these epidemiological observations support the theory of the so-called "gut-brain axis”, in which the gastrointestinal alterations contribute to the development of CNS diseases, possibly through inflammatory mechanisms.

### 2.2. Experimental Evidences of Gut Brain-Axis

Besides the epidemiological observations, numerous experimental studies, using animal models of UC and CD, have demonstrated that gastrointestinal inflammation may affect the brain. In these experimental models, the chemical agent 2,4,6- trinitrobenzene sulfonic acid (TNBS) or dextran sodium sulfate (DSS) were used to induce local, human IBD-like inflammation in the intestine of the mice [27,28]. These data demonstrated that intestinal inflammatory processes induce PD and AD-related pathological changes in the brain, including increased BBB permeability, neuro-inflammation, α-syn aggregation, and dopaminergic loss [29,30,31]. 

These experimental results are in accordance with the epidemiological observations suggesting that intestinal inflammation can induce PD and AD-associated pathological alterations in the CNS. The relevant experimental evidences demonstrating the GBA crosstalk were summarized in Table 1. 

## 3. Role of Intestinal Dysbiosis and Inflammation in CNS Diseases 

In the healthy intestine, the mucosal barrier formed by epithelial cells and the junctional complexes, including tight junctions (TJ), adherent junctions, desmosomes, and gap junctions between the epithelial cells, play a pivotal role in the separation of the host’s luminal microbes from the gastrointestinal immune system [46]. In IBD, the composition of the microbiome and their metabolite production significantly changes and has been suggested as a causative factor that induces pathological alterations in the intestine [47].

The gut microbiome in healthy adults is mainly composed by two dominant bacterial phyla, Firmicutes (F) and Bacteroidetes (B), that represent more than 90% of the intestinal community, and also by other less dominant phyla, including Proteobacteria, Actinobacteria, and Verrucomicrobia [48]. The F/B ratio is considered as an important index of the health of the gut microbiome [49]. An imbalance in the composition of the microbiome is commonly found in IBD patients, with a trend toward the reduction of beneficial bacteria and an increase in invasive, toxin-producing, mucosal adherent bacterial species such as *Escherichia coli* (*E. coli*) [50,51]. *E. coli* strains impair the intestinal epithelial barrier in different ways, through a direct effect of their toxic factors that damage the intestinal actin cytoskeleton and via disruption of the epithelial barrier. They also impair it by adhering to the epithelial layer, thus stimulating the production of pro-inflammatory cytokines and disrupting the epithelial tight junctions and the intestinal barrier [51].

Dysbiosis of the gut can also lead to changes in the production of microbial metabolites. Indeed, it has been demonstrated that the metabolic profile is different between healthy and IBD patients [52]. The metabolites in the gut are actively absorbed and affect the integrity of the intestinal epithelial layer. Indeed, short-chain fatty acids such as acetate, propionate, and butyrate were demonstrated to facilitate the turnover of intestinal epithelial cells, thereby protecting intestinal integrity [53]. In addition, IBD patients tend to have lower fecal levels of short-chain fatty acids [53,54]. On the other hand, there are metabolites, such as the sulfur-containing metabolites hydrogen sulfide or bile acid chenodeoxycholic acid, that damage intestinal epithelial cells and induce mucosal inflammation [52,55].

The injured epithelial cell layer allows the bacteria to penetrate into the mucosal layer and activate the local immune cells, including dendritic cells and macrophages, the frontline cells of innate immunity. These immune cells express a wide range of pattern recognition receptors, including Toll-like receptors (TLR), allowing them to recognize and respond to a wide range of endogenous substances and exogenous pathogens [46,56]. The activation of TLRs initiate signal transduction pathways that culminate in the activation of nuclear factor kappa B (NF-κB), interferon regulatory factors, or mitogen-activated protein kinases to regulate the expression of cytokines, chemokines, and type I interferons, reactive oxygen species that ultimately protect the host from microbial infection [57]. In IBD the immune cell activation in the intestinal wall contributes to the tissue damage by enhancing the production of matrix metalloproteinases (MMP) responsible for intestinal ulceration [58,59]. In the injured intestine the above-mentioned inflammatory mediators can enter into the microcirculation of the gastrointestinal wall, thereby spreading throughout the body and making the inflammation systemic (Figure 1). 

Systemic inflammation has been demonstrated to induce pathological changes in the BBB integrity. Indeed, increased permeability of BBB has been described in various human diseases associated with systemic inflammation, including sepsis, diabetes, multiple sclerosis, or obesity [60,61,62,63]. The pro-inflammatory factors in the circulation have been suggested to play a key role in the disruption of the molecular connection between the capillary endothelial cells of BBB [62]. Indeed, bacterial lipopolysaccharide (LPS), IL-1β, IL-6, and TNF-α inhibit the transcription of junction proteins and induce the cytoskeleton-mediated redistribution of TJs from the cell membrane to the cytoplasm, resulting in junctional disorganization and the separation of the cells from each other [62,64]. Both experimental and human studies demonstrated that intestinal pathologies are associated with increased levels of inflammatory mediators, including IL-6, TNF-α, bacteria, and LPS in the circulation. [36,43,65,66,67,68,69,70,71]. Moreover, the above-mentioned data experiments using TNBS or DSS induced mice model of IBD demonstrated that intestinal inflammation leads to decreased expression of the TJ protein occludin and claudin-5 in the brain and increases the permeability of the BBB (Table 1). Therefore, it can be reasonably assumed that systemic inflammation in IBD may contribute to BBB impairment.

The role of the tight junction protein zonulin was also suggested in the impairment of BBB integrity [72]. Indeed, as a result of intestinal inflammation triggered by gluten or microbial imbalance, epithelial zonulin is released from its junctional complexes into the circulation [73]. Circulatory zonulin can bind to epidermal growth factor receptors and protease-activated receptor 2 receptors on the endothelial cells of BBB, resulting in dysfunctional reorganization of endothelial TJs [74]. Indeed, the study of Rahman et al. demonstrated that zonulin can modify the cellular localization of zonula occludens 1 (ZO-1), claudin-5, and occludin, thereby influencing the permeability of BBB [72]. 

Injury of the BBB is of great importance in neuroinflammation since it paves the way for intestine-derived exogenous and endogenous substances to enter into the brain and bind to their receptors on glial cells (astrocyte and microglia), thereby triggering their activation. Activated glial cells are the major source of those inflammatory cytokines, chemokines, and reactive oxygen species (ROS) which play a central role in the development of neurodegenerative disorders [75,76,77].

It is important to note that the vagus nerve-mediated crosstalk between gut and brain may also play a role in GBA, however, its role is still controversial. Previously, Holmqvist et al. found that the pathologic form of α-syn from a PD patient’s brain lysate injected into the gastric wall of rodents is taken up and transported from the gut to the dorsal motor nucleus via the VN [5]. However, on the other hand, stimulation of VN was demonstrated to inhibit cytokine production of the gut, including that of TNF-α, and improve intestinal integrity [78].

Taken together, an increasing number of experiments suggest that the crosstalk between gut and brain can be mediated by intestine-derived systemic inflammation that disrupts the BBB, allowing the spreading of harmful molecules into the brain, thereby facilitating the development of neurodegenerative diseases.

## 4. PARK7/DJ-1

PARK7/DJ-1 is an evolutionarily conserved 20 kDa protein that forms homodimers and is composed of 189 amino acids. PARK7/DJ-1 has 3 cysteine residues, of which Cys106 is perhaps the most studied, with four possible oxidative statuses (Cys106-SH/-SOH/-SO_2_H/-SO_3_H) that affects its three-dimensional structure and thereby its functions [79,80,81]. Excessive oxidation of PARK7/DJ-1 is characterized by the formation of Cys106-SO_3_H and results in the inactivation and degradation of the protein [80,81]. The relevance of this property of PARK7/DJ-1 is well-demonstrated by the studies of Kitamura et al., which prove that pharmacological prevention of the excessive oxidation of Cys106 of PARK7/DJ-1 is protective against oxidative stress-induced neuronal cell death [8,82].

PARK7/DJ-1 is expressed in almost all, if not all, human cells [83]; it is primarily localized in the cytoplasm, however, its occurrence in the mitochondria [84] and nucleus [85] and also its secretion into the extracellular space was reported [86]. Previously, the expression of PARK7/DJ-1 was mainly investigated regarding malignant tumors and neurodegenerative diseases. In malignant diseases, the increased expression of PARK7/DJ-1 was demonstrated in glioblastoma, non-small cell lung, thyroid, breast, hepatocellular, and colorectal carcinoma [87]. In addition, the elevated PARK7/DJ-1 expression was closely correlated with the poor survival of patients with colorectal and pancreas cancers [87,88,89]. On the contrary, the reduced expression and/or the increased amount of dysfunctional overoxidized form of PARK7/DJ-1 was found in the brain of patients with various neurodegenerative disorders, including PD, AD, Huntington’s disease, and ischemic stroke [90]. 

### 4.1. Regulation of PARK7/DJ-1

Most of the studies suggest that the main determinant of the proteomic degradation of PARK7/DJ-1 is oxidative stress [80,81]. However, an increasing amount of data shows that regulatory mechanisms leading to decreased PARK7/DJ-1 expression or function are more complex [87]. Indeed, it has been shown that the cellular stress-activated tumor suppressor p53 pathway leads to decreased expression of PARK7/DJ-1 in mouse embryonic fibroblasts cells [91]. The results of Qin et al. demonstrated that the Bcl-2 associated athanogene 5 (BAG5) chaperon interacts with PARK7/DJ-1 and negatively regulates its dimerization, thereby facilitating its degradation under oxidative stress in human embryonal kidney (HEK)-293 cells [92]. The role of MMP-3 was also suggested in the proteomic cleavage and degradation of PARK7/DJ-1 in CATH neuronal cells [93]. Our very recent results showed that inflammatory mediators of IBD, including LPS, TNF-α, and TGF-β, decreased the expression of PARK7/DJ-1 in HT-29 colonic adenocarcinoma cells [11]. 

The micro RNA (miR) mediated regulation of PARK7/DJ-1 expression has also been shown. Indeed, it was recently demonstrated that the miR-128-3p directly reduces the expression of PARK7/DJ-1 in hepatocellular carcinoma cells by binding to the 3′UTR region of PARK7/DJ-1 [94]. Similarly, miR-203 was shown to reduce PARK7/DJ-1 expression in pancreatic cancer cells [95]. Furthermore, the overexpression of miR-494 decreased the PARK7/DJ-1 level of adipocytes and neuroblastoma cells resulting in increased vulnerability of cells to oxidative stress. In line with this data, in an MPTP-induced mouse model of PD, the overexpression of miR-494 negatively regulated PARK7/DJ-1 levels and exacerbated neurodegeneration [96]. 

Besides the negative regulators of PARK7/DJ-1, recent studies identified molecules that can increase PARK7DJ-1 expression and functions. Indeed, it was shown that the signal transducer and activator of transcription factor (STAT5A) increases the expression of PARK7/DJ-1 in leukemic pre-B cells, thereby preventing their apoptotic cell death [97]. The role of cellular homeostasis regulator cell cycle autoantigen (SG2NA) has also been demonstrated to increase the expression of PARK7/DJ-1 by inhibiting its proteasome-mediated degradation in Neuro2a neuroblastoma cells [98,99,100]. Finally, our recent study showed that IL-17 and hydrogen peroxide increases the expression of PARK7/DJ-1 [11]. Interestingly, while increased oxidative stress is one of the main mediators of proteomic degradation of PARK7/DJ-1, our results showed that as a positive feedback mechanism it can facilitate its synthesis [11].

Based on the currently available data, the PARK7/DJ-1 level is negatively regulated by processes associated with inflammation and oxidative stress. However, at the same time, harmful effectors can increase PARK7 expression, which may help the survival of the cells (Table 2). 

### 4.2. Functions of PARK7/DJ-1

The importance of PARK7/DJ-1 in cellular stress response and survival was first demonstrated by the fact that its loss-of-function mutation results in increased neuronal cell death by reducing resistance against oxidative stress, causing early onset of neurodegenerative diseases, including juvenile PD [7,101,102]. Many functions of PARK7/DJ-1 play a key role in the protection of cells against oxidative stress. Indeed, PARK7/DJ-1 is a peroxiredoxin-like peroxidase, which can directly eliminate intracellular reactive oxygen species (ROS) through the oxidation of its Cys-106 [103]. Moreover, by binding to complex I of the mitochondrial oxidative phosphorylation system PARK7/DJ-1 may play a role in the maintenance of the mitochondrial respiratory chain, thereby controlling the generation of intracellular ROSs [104,105].

However, in terms of antioxidant protection, the most important function of PARK7/DJ-1 is the regulation of transcription factors that are crucial in the cellular defense mechanisms against oxidative stress. Indeed, PARK7/DJ-1 activates the nuclear factor erythroid 2–related factor 2 (NRF2), which is one of the main transcriptional regulators of antioxidant response genes, including thioredoxin (TRX1), glutamate-cysteine ligase catalytic subunit (GCLC), heme oxygenase-1 (HMOX1), NAD(P)H quinone dehydrogenase-1 (NQO1) [106,107].

In addition to its direct antioxidant role, PARK7/DJ-1 has been demonstrated to influence the degradation of the advanced glycation end products (AGE) (Figure 2). AGE formation is facilitated by reactive glucose degradation products (GDP), such as 3-deoxyglucosone, 3,4-dideoxyglucosone-3-ene, glyoxal, or methylglyoxal, originated from the non-enzymatic, spontaneous decomposition of reducing monosaccharides [108]. AGEs then lead to the induction of oxidative stress and inflammation through the activation of their AGE receptors (RAGE). AGEs play a central role in the pathomechanism of several diseases, including IBD, AD, PD, stroke, alcoholic brain damage, or diabetes mellitus [109,110]. To eliminate toxic GDPs and AGEs, the glutathione system and different proteases degrade the GDPs leading to the generation of nontoxic lactate. PARK7/DJ-1 is involved in this process at several points. On the one hand, it was demonstrated that PARK7/DJ-1 upregulates the glutathione (GSH) synthesis, which is an essential cofactor of glyoxalase (Glo1)-1, and 2, which are responsible for the elimination of GDPs (Figure 1) [111,112,113]. During this enzymatic process, the deteriorative GDPs will be degraded to bioavailable pyruvate through lactoylglutathione and lactate intermediate products. Furthermore, PARK7/DJ-1 has its glyoxalase and deglycase activities [114,115]. PARK7/DJ-1, due to its glyoxalase activity, prevents the glycation of macromolecules, including the free amino residues of proteins, lipids, and DNA by cleaving the GDPs (Figure 1). In addition, as a deglycase, PARK7/DJ-1 repairs glycated proteins by releasing lactate from the damaged amino-acid residues (Figure 2) [116,117].

Moreover, under oxidative stress, PARK7/DJ-1 mediates cell survival by activating Ras-dependent extracellular signal-regulated kinase (ERK1/2) and apoptosis signal-regulating kinase 1 (ASK-1) pathways [118,119].

As a molecular chaperone PARK7/DJ-1 has been demonstrated to prevent the aggregation of different macromolecules, including α-syn, which is a key factor in the development of PD [120,121].

Finally, the role of PARK7/DJ-1 in the inflammatory mechanism was also demonstrated; however, its anti- and pro-inflammatory role is still controversial and seems disease-specific. Indeed, anti-inflammatory properties of PARK7/DJ-1 have been suggested by Kim et al. who found that astrocytes and microglia originated from PARK7/DJ-1-knockout (KO) mice exhibited increased expression of inflammatory mediators, including TNF-α, Cyclooxygenase-2 (COX-2), and (inducible nitric oxide synthase) iNOS in response to interferon-gamma (IFN-γ) treatment [122]. Similarly, Peng et al. demonstrated that RNA interference-mediated PARK7/DJ-1 silencing resulted in increased IL-1ß, IL-6, and TNF-α expression in the cerebral cortex of rats after I/R injury and cultured brain astrocytes as well [123]. Moreover, our previous study investigating the role of PARK7/DJ-1 in DSS-induced colitis also demonstrated that pharmacological activation of PARK7/DJ-1 resulted in the decreased expression of pro-inflammatory cytokines, including IL-1β and IL-6, and decreased diseases activity index in the DSS treated mice [11]. However, other studies suggested the pro-inflammatory role of PARK7/DJ-1. Chen et al. demonstrated that RNA interference-mediated PARK7/DJ-1 silencing leads to decreased IL-6 and TNF-α expression in activated macrophages [124]. Similarly, Kim et al. found that PARK7/DJ-1 deficiency resulted in the reduced expression of IL-1β and IL-6 in adipocytes and decreased the activity of pro-inflammatory transcription factor NF-κB in macrophages [125].

Taken together, the most widely demonstrated function of PARK7/DJ-1 is its anti-oxidant functions. However, numerous additional functions of PARK7/DJ-1 were identified, including its enzyme activities, anti-apoptotic effect, and regulatory role on inflammation, respectively.

## 5. Role of PARK7/dj-1 in the Pathogenesis of Neurodegenerative Diseases

The connection between PARK7/DJ-1 and neurodegenerative diseases was first suggested nearly two decades ago when it was identified as a causative factor in rare inherited forms of PD [7]. Since then, the role of PARK7/DJ-1 in neurodegenerative diseases has been intensively studied, and it has become clear that PARK7/DJ-1 may play a role not only in PD but in almost all neurological diseases associated with oxidative stress, inflammation, and tissue damage [9,126,127].

### 5.1. Genetic Evidence for the Role of PARK7/DJ-1 in Parkinson’s Disease

The group of Parkinson’s disease molecules involves 23 genes located on different chromosomes and having different functions [128]. However, one thing that is common in the members of the PARK family is that they are all associated with the higher risk of PD [129]. This relationship was first suggested in 2003 when Bonifati and colleagues found a large (about 14 kb) deletion and a missense mutation (Leucine166Proline, L166P) in the PARK7/DJ-1 gene in a Dutch and an Italian family, leading to the identification of PARK7/DJ-1 as a causative gene for familial PD with recessive inheritance [7]. Since then, more than 20 PARK7/DJ-1 mutations have been associated with PD [130].

### 5.2. Role of PARK7/DJ-1 in Parkinson’s Disease

Experimental data suggest that PARK7/DJ-1 plays a protective role in neurodegenerative diseases via its antioxidant properties. Indeed, the lack of PARK7/DJ-1 in stem cell-derived neurons and SH-SY5Y cells resulted in increased vulnerability to oxidative stress [131,132,133]. In addition, the neuroprotective effect of the administration of recombinant PARK7/DJ-1 was demonstrated in the rodent model of 6-hydroxydopamine (6-OHDA) and MG-132 treatment-induced PD [134]. Furthermore, it has been demonstrated that pharmacological protection of PARK7/DJ-1 against overoxidation preserves its antioxidant properties. Indeed, Miyazaki et al. and Kitamura et al. identified small molecule compounds, including UCP0054277, UCP0054278, and Compound 23 (Comp23), that can bind to the C106 region of PARK7/DJ-1 and keep it in reduced, biologically active form [8,135]. The protective effects of these compounds against oxidative stress were confirmed in hydrogen peroxide- (H_2_O_2_) treated wild type and PARK7/DJ-1-knockdown SH-SY5Y neuronal cells [8,135,136]. In further experiments, they also demonstrated that administration of UCP0054278 and or Comp23 suppressed the loss of dopaminergic neurons and motor dysfunction in an animal model of 6-OHDA or rotenone-induced PD [8,126]. Glyoxalase activity of PARK7/DJ-1 in neuroprotection may also be of great importance since AGEs have been suggested to contribute to the development of neurodegenerative diseases. Indeed, glycation-mediated AGE formation has been reported in the Lewy bodies in PD patients [137]. The relationship between AGEs and PD could be due to the ability of AGEs to cross-link α-syn, as has been shown using in vitro studies [138].

### 5.3. Role of PARK7/DJ-1 in Alzheimer’s Disease

The role of PARK7/DJ-1 has also been suggested in AD. It was shown that the PARK7/DJ-1 binding compound UCP0054278 improved the AD-related cognitive deficits and prevented the degeneration of synaptic functions in AD modeling APdE9 transgenic mice [9]. Similarly, a recent study by Cheng et al. demonstrated that overexpression of *DJ* in the brain by lentiviral infection ameliorated β-amyloid protein (Aβ) deposition and the cognitive function of 5XFAD transgenic mice modeling AD [139]. The results also demonstrated that reactive oxygen species and oxidative stress marker malondialdehyde content were significantly decreased, while the antioxidant superoxide dismutase activity was significantly increased in the brain of 5XFAD mice overexpressing PARK7/DJ-1 [139]. In addition, AGEs are present in amyloid plaques in the brain of AD patients and have been suggested to promote the aggregation of Aβ and tau [140]. According to experimental data, the glyoxalase activity of PARK7/DJ-1 may reduce these deleterious effects of AGEs in neurons. Indeed, the protective role of PARK7/DJ-1 against dicarbonyl stress was demonstrated using mouse embryonic fibroblasts, human SH-SY5Y cells, and *C. elegans*, as well [114].

### 5.4. Role of PARK7/DJ-1 in Huntington’s Disease

Huntington’s disease (HD) is an autosomal dominant inherited disease associated with polyglutamine expansion in the huntingtin (Htt) protein, leading to its misfolding and toxic aggregation [141]. A recent study by Sajjad et al. demonstrated that the level of oxidized PARK7/DJ-1 Cys106 level was elevated in the frontal cortex of HD patients [127]. They demonstrated that overexpression of PARK7/DJ-1 ameliorated mutant Htt toxicity in a yeast and Drosophila model of HD, suggesting the importance of the chaperoning activity of PARK7/DJ-1 in vivo. Their results also demonstrated that mild oxidation of PARK7/DJ-1 at cysteine 106 is required for its chaperone function; however, the complete oxidation of cysteine 106 leads to impaired PARK7/DJ-1 and detrimental cellular outcomes [127].

### 5.5. Role of PARK7/DJ-1 in Ischemia-Reperfusion Induced Brain Injury

The importance of PARK7/DJ-1 has also been demonstrated in the ischemic-reperfusion injury of the brain. Indeed, intrastriatal injection of recombinant human PARK7/DJ-1 markedly reduced infarct size after middle cerebral artery occlusion of rats and protected SH-SY5Y against H_2_O_2_-induced apoptosis [133]. PARK7/DJ-1-deficient animals produced a significantly larger infarct size in the animal model of Endothelin-1 induced stroke compared to wild-type controls [142]. On the contrary, the administration of ND13 corresponding to the 13-N-terminal amino acids of PARK7/DJ-1 was shown to improve motor function after ischemic injury [143]. Similarly, PARK7/DJ-1 binding Comp23 reduced the infarct size of cerebral ischemia in rats [8,136,144].

## 6. Role of PARK7/DJ-1 in the Pathogenesis of Gastrointestinal Diseases

In addition to the previously discussed effects of PARK7/DJ-1 in the pathomechanism of CNS diseases, its role in the disease of other organs including the heart [145,146,147], lung [148,149] and intestine [10,12,13,14,114,150,151,152] was recently studied. However, in the present section, we focus on the recently emerged role of PARK7/DJ-1 in the pathomechanism of intestinal diseases.

### 6.1. Genetic Evidence of the Role of PARK7/DJ-1 in Gastrointestinal Diseases

First, the genome-wide association (GWA) study of Dubois et al. involving 4533 coeliac disease cases and 10750 controls suggested that the genomic region of the short arm of chromosome 1 containing the PARK7/DJ-1 gene and also that of TNF Receptor Superfamily Member 9 (TNFRSF9) is strongly associated with the risk of coeliac disease [153]. A year later, in 2011, another GWA study comprising 6687 cases with ulcerative colitis (UC) and 19718 controls prepared by Anderson et al. revealed that the 1p36 chromosomal region, containing TNFRSF9, ERFF11, UTS2, and PARK7/DJ-1 genes, is associated with a higher risk of UC [150].

Not long after this, Lee et al. demonstrated that cDJR-1.1, the *C. elegans* homolog of the human PARK7/DJ-1 is expressed in the intestine of the worms. In addition, their results showed that lack of cDJR-1.1 makes the worms vulnerable to glyoxal-induced intestinal toxicity, giving the first in vivo evidence suggesting the protective role of PARK7/DJ-1 in intestinal pathology [114].

### 6.2. Role of PARK7/DJ-1 in Coeliac Disease

The first direct human evidence suggesting the possible role of PARK7/DJ-1 in the pathomechanism of small intestinal diseases was the study of Vörös et al. [10]. In this study, our research group demonstrated the increased mRNA expression and protein level of PARK7/DJ-1 in the small intestinal mucosa of patients with untreated coeliac disease. In this study, we found that PARK7/DJ-1 immunopositivity is present in the epithelial cells of the duodenal crypt, and also in the lamina propria of duodenal biopsies derived from therapy-naive children with celiac disease. Moreover, we found that, following the introduction of a gluten-free diet, the amount of PARK7/DJ-1 normalizes, suggesting the possible role of PARK7/DJ-1 in the pathomechanism of celiac disease.

More recently, our research group examined the possible function of PARK7/DJ-1 on the pathomechanism of intestinal inflammation in more depth [12]. In this study, Comp-23, a PARK7/DJ-1 binding compound that protects PARK7/DJ-1 from overoxidation, was used to investigate the functional role of PARK7/DJ-1 in the pathomechanism of celiac disease [8].

We found that Comp23 decreased the intracellular accumulation of ROS in the H_2_O_2_ treated duodenal FHs74Int, epithelial cells. Our finding was in line with previous experiments demonstrating that lack of PARK7/DJ-1 is associated with increased accumulation of ROS in different cells, including dopaminergic cells, mast cells, regulatory T cells, or skeletal muscle cells, suggesting the role of PARK7/DJ-1 in antioxidant defense [154,155,156]. Investigating the underlying molecular mechanism, we found that Comp23 treatment increases the expression of NRF2 transcription factor—one of the major regulators of antioxidant defense—and also the expression of several NRF2 dependent antioxidant genes, including TRX1, GCLC, HMOX1, and NQO1, in the FHs74Int cells exposed to oxidative stress [12,157]. Indeed, our results are in line with the literature suggesting that PARK7/DJ-1 may influence the activation of the NRF2-related antioxidant genes and thereby oxidative stress [158]. Not surprisingly, by the increased antioxidant preparedness of the Comp23 treated cells, we demonstrated that Comp23 treatment improves the survival of H_2_O_2_-treated FHs74Int cells. Investigating the molecular mechanism of PARK7/DJ-1 on cell viability, we found that Comp23 treatment increased the expression of transcription factor tumor antigen P53 (TP53) and that of its target genes, including proliferating cell nuclear antigen (PCNA), cyclin-dependent kinase inhibitor 1-p21 (CDKN1A), apoptosis regulator Bcl-2 (BCL2), and apoptosis regulator BAX (BAX) in the H_2_O_2_-treated FHs74Int cells [12].

Finally, we have investigated the role of PARK7/DJ-1 in the maintenance of the small intestinal mucosa integrity. We demonstrated that Comp23 treatment restores the expression of cell adhesion molecules (CAM), including ZO-1, cadherin 1 (CDH1), vinculin (VCL), and integrin Subunit Beta 5 (ITGB5), and the healthy architecture of the actin cytoskeleton-CAM complexes in the FHs74Int cells exposed to oxidative stress (Figure 3). Not surprisingly, in our experiments, Comp-23 treatment also restored the oxidative stress-induced permeability of small intestinal sacs ex vivo, suggesting the role of PARK7/DJ-1 in the control of oxidative stress and the maintenance of mucosal integrity.

### 6.3. Role of PARK7/DJ-1 in Inflammatory Bowel Disease

Despite the growing interest, there are relatively little data about the role of PARK7/DJ-1 in the pathomechanism of IBD. Recently, Di Narzo et al. investigated the plasma proteome of adult patients with Crohn’s disease (CD; *n* = 126) and ulcerative colitis (UC; *n* = 46) compared to that of healthy subjects (*n* = 72) using a high-throughput SOMAmer-based capture array [152]. They found that a total of 493 proteins showed altered levels in the plasma of patients with IBD compared to healthy subjects; among them, 219 were up- and 274 were down-regulated. One identified protein with an increased presence in the plasma of UC patients compared to that of healthy subjects was PARK7/DJ-1.

More recently, Zhang et al. investigated the role of PARK7/DJ-1 in the pathomechanism of IBD [13]. They found decreased levels of PARK7/DJ-1 in the intestine of patients with active CD (*n* = 63) or UC (*n* = 23) compared to the healthy subjects (*n* = 15). They noted that the amount of PARK7/DJ-1 was even lower in the inflamed, compared to the non-inflamed mucosa of the individual patients. Interestingly, the results of our research group somehow differ from the study of Zhang et al. [11]. Indeed, we demonstrated an increased amount of PARK7/DJ-1 in the epithelial and lamina propria cells of macroscopically inflamed and non-inflamed mucosa of therapy-naive pediatric patients with CD compared to that of controls. However, in our study, the amount of PARK7/DJ-1 was at the level of control in the mucosa of children with UC.

In connection with the obvious contradictions of the aforementioned studies, it should be noted that these results are hardly comparable because of the different types of samples, patients, and methods. While Di Narzo et al. investigated the plasma proteome of adult IBD patients, our group determined the mucosal mRNA expression and protein level of PARK7/DJ-1 from fresh frozen biopsies of therapy-naive children. In fact, Di Narzo et al. also note that the plasma level of the investigated proteins had only little correlation with their expression in the intestine. Unfortunately, less is known about the paraffin-embedded colon sections used in the study of Zhang et al. Although they used a high number of samples, the detailed description of the samples and the patients remains unknown. Therefore, it is easy to suppose that the difference is mainly due to the origin of the samples, age, or medication of the patients.

Zhang et al. and our research group also investigated the expression of PARK7/DJ-1 in the colon of DSS-treated mice. According to these studies, the amount of PARK7/DJ-1 is increased on the 3^rd^ day of DSS treatment, however later on the seventh day of the experiment it decreases to the level of the untreated control animals or below [11,13].

Zhang et al. demonstrated that following DSS treatment, more serious clinical symptoms, such as decreased body weight and higher disease activity index (DAI), develop in the PARK7/DJ-1 KO compared to the WT mice. Accordingly, they demonstrated the increased expression of TNF-α, IL-1β, IL-6, IL-8, monocyte chemoattractant protein 1 (MCP-1), and C-C Motif Chemokine Ligand 3 (CCL3) in the colon of DSS treated PARK7/DJ-1 KO, compared to that of WT mice, suggesting the anti-inflammatory role of PARK7/DJ-1 (Figure 3).

Similarly, using the same DSS induced mouse model of colitis, we demonstrated that treatment of the mice with the PARK7/DJ-1 binging Comp23 improves the clinical symptoms, decreases the histological lesions, spleen enlargement, and the mucosal expression of IL-1β, IL-6, and IL10, and TGF-β of DSS-treated mice, suggesting the strong anti-inflammatory role of PARK7/DJ-1 and Comp23 (Figure 4).

The above in vivo studies were confirmed by in vitro experiments demonstrating the strong, bidirectional interaction between the inflammatory factors and PARK7/DJ-1. Indeed, we showed that while TNF-α, TGF-β, or LPS treatment inhibits, IL-17 or H_2_O_2_ treatment increases the synthesis of PARK7/DJ-1 in vitro. Moreover, our group and Zhang et al. demonstrated that the gene silencing of PARK7/DJ-1 increases the expression of TNF-α and IL-8 of HT-29 or Caco2 colon epithelial cells and decreases the expression of IL-1β and IL-6 (Figure 5).

To further specify the role of PARK7/DJ-1, Zhang et al. generated PARK7/DJ-1 bone marrow chimera mice. When PARK7/DJ-1 was disrupted only in the myeloid cells of the mice (KO to WT bone marrow chimera mice), the DSS treated animals had higher body weight and lower DAI compared to that control WT to WT chimera mice. Similarly, less pronounced activation of the NF-κβ signaling pathway and decreased expression of inflammatory cytokines, including IL-1β, IL-6, MCP1, and CCL3 were observed in the colon of KO to WT compared to WT chimera mice with DSS colitis (Figure 6). These observations suggest that the presence of PARK7/DJ-1 in immune cells may contribute to the exacerbation of DSS induced colitis, which is somehow in contradiction with the above experiment of the research group, demonstrating more serious symptoms and inflammation in the colon of PARK7/DJ-1 KO compared to WT mice.

When only the myeloid cells of the mice expressed PARK7/DJ-1 (WT to KO bone marrow chimera mice), the symptoms of the mice were more severe compared to that of KO to KO chimera mice. Also, the expression of inflammatory cytokines, including IL-1β, IL-6, MCP1, and CCL3 were higher in the colon of DSS treated WT to KO compared to KO mice with colitis, revealing that the presence of PARK7/DJ-1 in the immune cells may contribute to DSS-induced inflammation, especially in a colon lacking PARK7/DJ-1 (Figure 6). Although the interpretation of the above in vivo experiments, using bone marrow chimera mice, is quite challenging because of the lack of a common group of animals, Zhang et al. suggest that the lack of PARK7/DJ-1 in the epithelium is critical to improving colitis whereas the role of PARK7/DJ-1 in immune cells may be the reverse.

Taken together, the above studies clearly demonstrate the role of PARK7/DJ-1 in coeliac disease and IBD-related intestinal inflammation, however, further studies are needed to resolve the contradictions and elucidate the precise role of PARK7/DJ-1 in IBD.

### 6.4. The Role of PARK7/DJ-1 in Intestinal Dysbiosis

A recent publication of Singh et al. investigated the effect of PARK7/DJ-1 deficiency on the intestinal microbiome of healthy mice [14]. They investigated the effect of PARK7/DJ-1 deletion on bacterial composition of the intestine by 16S rRNA sequencing. Analysis of fecal samples showed that overall composition of the microbiome did not differ between PARK7/DJ-1−/− and PARK7/DJ-1+/+ and mice at the phylum level. However, calculation of the F/B ratio showed that it decreased significantly in PARK7/DJ-1−/− mice compared to PARK7/DJ-1+/+ mice, suggesting the functional role of PARK7/DJ-1 on the composition of the intestinal microbiome. The deeper analysis of the data showed an increased presence of Alistipes and Rikenella species in PARK7/DJ-1−/− mice. The role of these species has been described regarding the pathomechanism of IBD [159].

Changes in the composition of intestinal microbiome affect its metabolite production. Accordingly, they demonstrated that the amount of fecal and also that of serum amino acids, including valine, leucine, phenylalanine, alanine, tyrosine and isoleucine, were downregulated, whereas SCFAs, including malonate, dimethylamine, trimethylamine and acetoin, were upregulated in PARK7/DJ-1−/− mice compared to that of PARK7/DJ-1+/+ mice. Although the metabolic changes of PARK7/DJ-1−/− mice were complex, Singh et al. suggested that it can lead to metabolic stress of the intestine, as demonstrated by the increased inflammation of the intestine. Indeed, they found increased levels of pro-inflammatory monocyte chemotactic protein-1 (MCP-1) and calprotectin in feces of PARK7/DJ-1−/− mice compared to that of PARK7/DJ-1+/+ mice. However, it must be also noted that the expression of other pro-inflammatories such as granulocyte monocyte colony stimulating factor IFN-β decreased, and that of many others, including IL-12p70, TNF-α, IL-17A, IFN-γ, IL-23 and IL-6 did not change.

Finally, since the association of the intestinal microbiome and also that of PARK7/DJ-1 with neurodegenerative diseases is well known, they investigated the molecular biological changes in the midbrain of PARK7/DJ-1−/− mice by RNA-sequencing. They found the upregulation of PD related inflammatory genes, including polymerase family member 1 (Parp1) and MMP-8 in the midbrain of PARK7/DJ-1−/− mice compared to that of PARK7/DJ-1+/+ mice, suggesting that changes in the intestinal microbiome and/or lack of PARK7/DJ-1 may alter the pathology of the CNS.

## 7. Role of PARK7/DJ-1 in GBA Diseases

Based on our current knowledge, it is assumed that PARK7/DJ-1 represents a molecular link between intestinal and brain diseases. Indeed, recent studies demonstrated that PARK7/DJ-1, through its antioxidant and anti-inflammatory properties, plays a role in the maintenance of healthy intestinal microbiome and mucosal integrity, thus influencing the local and systemic inflammation characteristic for IBD.

As intestine-derived inflammatory factors can reach the BBB and impair its integrity, the inflammation can also spread also to the brain. The increased presence of inflammatory mediators induces the inflammation of CNS and may also alter the synthesis and function of PARK7/DJ-1 itself in the brain (Figure 7). Indeed, it has been shown that TNF-α, TGF-β, or LPS reduces the expression of PARK7/DJ-1, and also that increased amounts of MMP-3 may induce its degradation (Table 2). Also, the oxidative stress is an important factor that regulates the synthesis and function of PARK7/DJ-1 in the brain. Moreover, oxidative stress has been demonstrated to play a key role in the inactivation and degradation of PARK7/DJ-1 (Table 2).

PARK7/DJ-1 similarly to its effects in the gut has great importance in the protective mechanisms of the brain. Therefore, molecular mechanisms that alter PARK7/DJ-1 activity may contribute to the development of neurodegenerative disorders (Figure 7).

Although the function and regulation of PARK7/DJ-1 are still not completely understood, there is a growing body of evidence suggesting that it participates in the pathomechanism of diseases influenced by GBA.

## 8. Conclusions

There is a growing body of evidence about the connection between the development of gastrointestinal and neurodegenerative disorders. Epidemiological and experimental studies demonstrated that gastrointestinal diseases represent an increased risk of the development of neurodegenerative disorders, including Parkinson’s or Alzheimer’s diseases. Inflammatory and neuronal pathways may link the pathomechanism of intestinal and CNS diseases.

PARK7/DJ-1 plays a key role in maintaining the homeostasis of the intestine and brain via its antioxidant, anti-inflammatory, glyoxalase, and chaperone activity. Moreover, it can be suggested that intestine-derived systemic inflammation impairs the function of PARK7/DJ-1 in the CNS, thereby increasing the risk of the development of neurodegenerative diseases.

Although further studies are needed to accurately explore the role of PARK7/DJ-1 in the pathologic crosstalk between the gut and CNS, our existing knowledge draws our attention to PARK7/DJ-1 as a potential therapeutic target molecule for the treatment of GBA diseases.

## Figures and Tables

**Figure 1 ijms-23-06626-f001:**
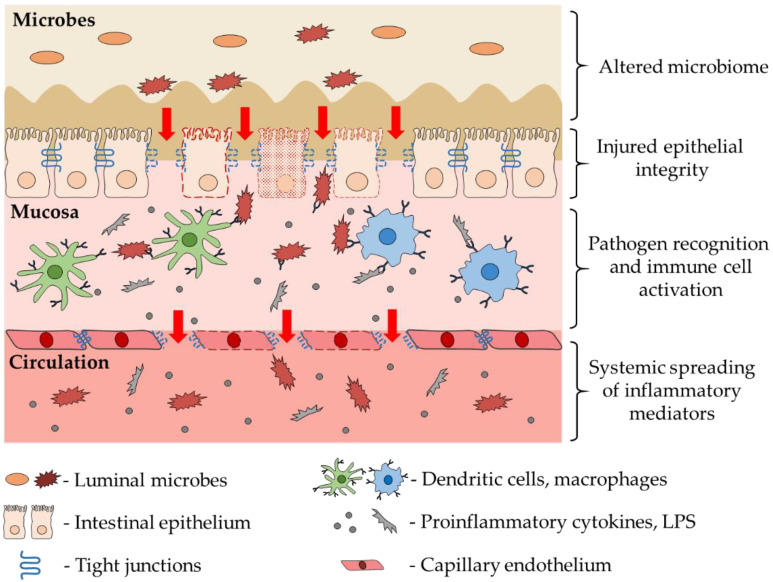
The role of intestinal alterations in the development of systemic inflammation. Disturbance of the gut microbiome leads to impairment of the epithelial barrier integrity and penetration of the luminal bacteria into the mucosal layer, thereby activating the resident immune cells. In the injured intestine the inflammatory factors, bacteria, and their parts can enter the circulation extending the intestinal inflammation to systemic inflammation.

**Figure 2 ijms-23-06626-f002:**
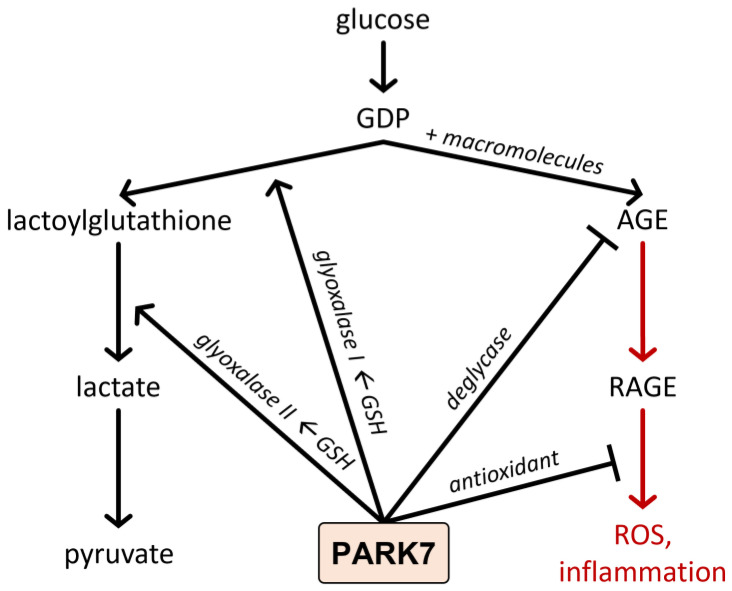
Effect of PARK7/DJ-1 on the formation of AGEs. AGE formation is facilitated by reactive GDPs. AGEs activate their AGE receptors leading to oxidative stress and inflammation. To eliminate toxic GDPs, the glutathione system degrades them to nontoxic lactate. PARK7/DJ-1 is involved in this process at several points. Indeed, PARK7/DJ-1 has its own glyoxalase activity, and upregulates the glyoxalase-1 and 2 enzyme activity, which is responsible for the elimination of GDPs. In addition, PARK7/DJ-1 has deglycase activity, thereby preventing the glycation of macromolecules and AGE formation. Finally, as an antioxidant, PARK7/DJ-1 protects the cells against the harmful effect of oxidative stress.

**Figure 3 ijms-23-06626-f003:**
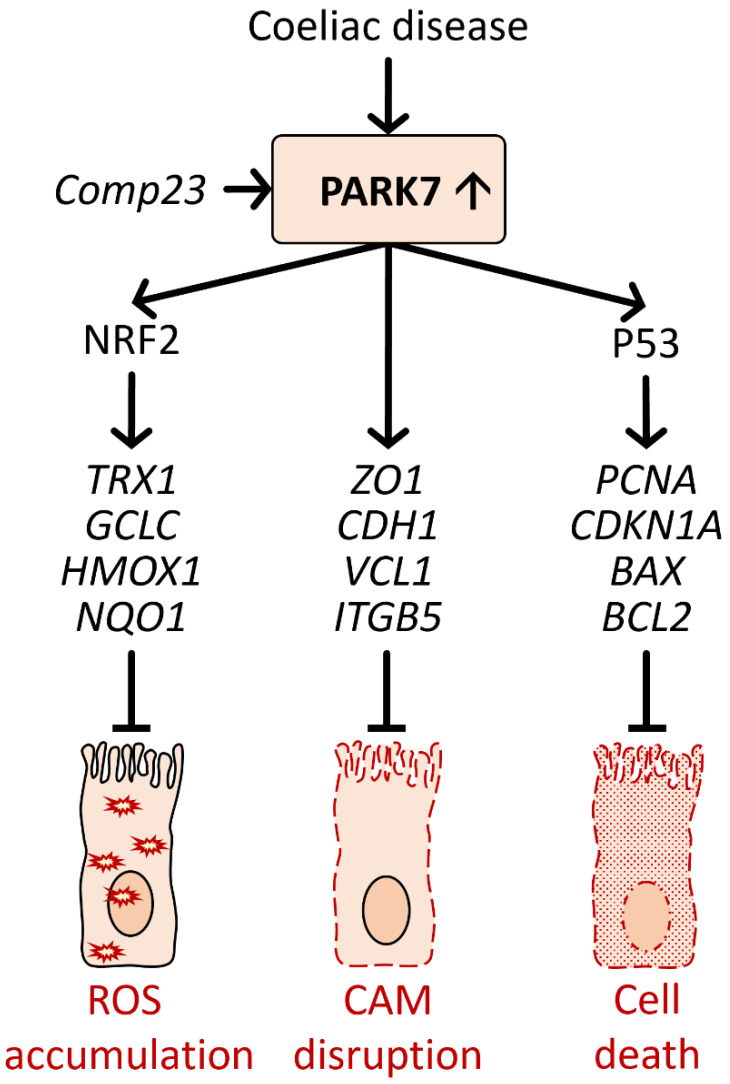
Role of PARK7/DJ-1 against oxidative damage of duodenal epithelial cells in celiac disease. Our study demonstrated that PARK7/DJ-1-binding Comp23 altered the expression of stress-response elements, including antioxidant and cell-cycle regulator genes, and normalized the expression and localization of CAMs, contributing to the maintenance of mucosal integrity as demonstrated by our ex vivo experiment.

**Figure 4 ijms-23-06626-f004:**
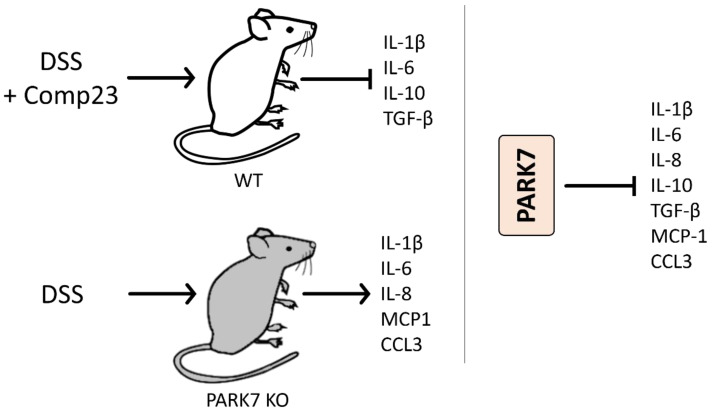
Effect of PARK7/DJ-1 on DSS treatment-induced colitis. Comp23 treatment decreased the mucosal expression of IL-1β, IL-6, IL-10, and TGF-β in DSS-treated mice. Similarly, following DSS treatment, decreased body weight, a higher disease activity index, and increased expression of TNF-α, IL-1β, IL-6, IL-8, MCP-1, and CCL3 were observed in the colon of DSS treated PARK7/DJ-1 KO compared to that of WT mice.

**Figure 5 ijms-23-06626-f005:**
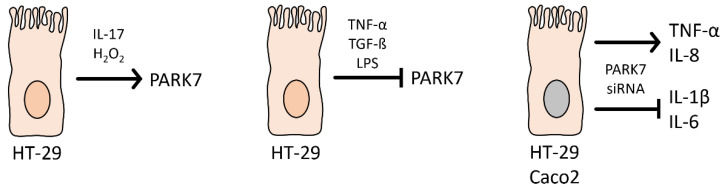
Effect of inflammatory factors on the synthesis of PARK7/DJ-1 and that of gene silencing of PARK7/DJ-1 on the synthesis of different inflammatory factors. IL-17 and H_2_O_2_ treatment decrease while TNF-α, TGF-β, and LPS treatment increases the expression of PARK7/DJ-1 in HT-29 colonic epithelial cells. PARK7/DJ-1 RNA silencing facilitates TNF-α and IL-8 expression and decreases IL-1β and IL-6 expression in HT-29 or Caco2 colon epithelial cells.

**Figure 6 ijms-23-06626-f006:**
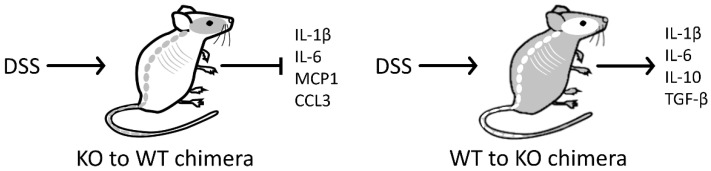
The role of PARK7/DJ-1 on immune cell activation in DSS-induced colitis. Symptoms of DSS-induced colitis were milder, and the expression of inflammatory cytokines was lower in the colon of KO to WT chimeric mice compared with WT to WT mice. Nevertheless, symptoms of DSS-induced colitis were more severe, and the expression of inflammatory cytokines was higher in the colon of WT to KO chimeric mice compared with KO to KO mice.

**Figure 7 ijms-23-06626-f007:**
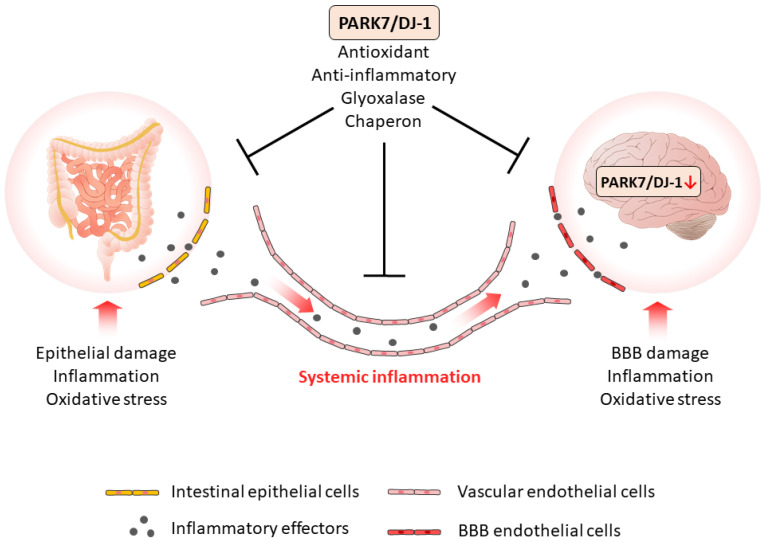
The role of PARK7/DJ-1 in GBA diseases. PARK7/DJ-1, through its antioxidant, anti-inflammatory, glyoxalase, and chaperon activity properties, plays a role in the maintenance of intestinal integrity, thus diminishing the local and the systemic inflammation. The systemic inflammation of intestinal origin can be considered as a possible factor that negatively regulates PARK7/DJ-1 in the brain. Similarly to the gut, PARK7/DJ-1 also plays a protective role in the brain, therefore, its negative regulation enhances pathological GBA crosstalk and the development of neurodegenerative diseases.

**Table 1 ijms-23-06626-t001:** Experimental results demonstrating the pathological crosstalk between the gut and brain.

Animal Model of IBD	Effect on the CNS	Refs.
TNBS-induced colitis in rabbit	Increased blood-brain barrier permeability	Hathaway et al. [32]
TNBS-induced colitis in rat	Elevated blood-brain barrier permeability and reduced endothelial barrier antigen expression	Natah et al. [33]
	Increased interleukin IL-6 expression in the hypothalamus and cerebral cortex	Wang et al. [34]
DSS-induced colitis in mouse	Elevated TNF-α, IL-1ß, and IL-6 expression in the substantia nigra	Villarán et al. [35]
	Increased TNF-α and IL-6 expression in the cortex and decreased TJ protein occludin and claudin-5 in the brain	Han et al. [36]
	Increased nigral level of IL-1ß and dopaminergic neuron death	Garrido Gil et al. [37]
	Increased COX-2 expression in the hippocampus and hypothalamus	Do et al. [38]
	α-syn aggregation in the midbrain	Grathwohl et al. [39]
	Microglial polarization into M1 and M2 phenotype in the medial prefrontal cortex	Sroor et al. [40]
	Increased IL-1ß, IL-6, TNF-α and IL-10 expression in the hippocampus	Gampierakis et al. [41]
	NLRP3 activation, amyloid plaque accumulation, and apoptosis in hippocampus, Cortex	He et al. [42]
	Elevated IL-1ß and TNF-α expression in the brain	Talley et al. [43]
	Increased microglia and astrocyte activation and loss of dopaminergic neurons in the substantia nigra pars compacta after PD inducing MPTP treatment	Gil-Martínez et al. [44]
	Increased neurotoxic effect of MPTP treatment	Houser et al. [45]

**Table 2 ijms-23-06626-t002:** Molecular regulation of PARK7/DJ-1.

Molecule	Effect on PARK7/DJ-1	Tissue or Cell Type	Refs.
Negative regulators of PARK7/DJ1			
H_2_O_2_	Overoxidation	Human brain	[80,81]
p53	Reduced expression	mouse embryonic fibroblasts	[91]
BAG5	Decreased stability	HEK293 human embryonic kidney	[92]
MMP-3	Proteomic fragmentation	CATH.a mouse neuronal	[93]
LPS	Reduced expression	HT-29 human colonic adenocarcinoma	[11]
TNF-α	Reduced expression	HT-29 human colonic adenocarcinoma	[11]
TGF-β	Reduced expression	HT-29 human colonic adenocarcinoma	[11]
miR-128-3p	Reduced expression	Human hepatocellular carcinoma	[94]
miR-494	Reduced expression	3T3-L1 mouse adipocytes and Neuro-2a neuroblastoma	[96]
miR-203	Reduced expression	SW1990/DDP human pancreatic cancer cells	[95]
Positive regulators of PARK7/DJ1			
STAT5A	Increased expression	human leukemic pre-B	[97]
SG2NA	Protection from degradation	Neuro2a neuroblastoma	[99,100]
IL-17	Increased expression	HT-29 human colonic adenocarcinoma	[11]
H_2_O_2_	Increased expression	HT-29 human colonic adenocarcinoma	[11]

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
