# Peer review of "PARK7/DJ-1 as a Therapeutic Target in Gut-Brain Axis Diseases"

_ijms, 2022, doi:10.3390/ijms23126626_

Round 1

Reviewer 1 Report

This review is interesting. Few improvements are suggested in below:

1- in  Epidemiological evidence of a gut-brain axis section, line 92-98, how do you connect this to the topic? maybe more clear connection is needed in this section

2- section related to Experimental evidences of gut brain-axis seems too short and most references from 2018, more related -new experimental evidence should be included. Making this as table includes the experimental evidence, how it connected to the topic, and the references will be easier for the reader to have an overview about the amount of related- experiments have been done. 

3- The link between the gut and brain: the role of inflammation line 122-145, support the text with figure or picture explaining the role of inflammation. 

4- lines 150-159 is not clear, better connection needed here.

5- ROLE OF PARK7/DJ-1 IN THE PATHOGENESIS OF NEURODEGENERATIVE DISEASES  line 322 supporting references is needed here. 

6- Conclusion should be improved. 

Reviewer 2 Report

Re: Manuscript ID: ijms-1728074

This is a well written review dealing with the intriguing role of PARK7/DJ-1 in the pathogenesis of gastrointestinal and neurodegenerative diseases. The authors have experimental expertise in this field. Anyway, some arguments need to be better developed and commented. The manuscript needs to be restyled. Some changes are suggested to improve the paper.

Points of criticism

The role of PARK7/DJ-1 in the pathogenesis of gastrointestinal and neurodegenerative diseases was well described, while the relationship between these two pathological conditions (gut-brain axis) was poorly examined. The authors mentioned the importance of the microbiome, but the specific role of the different bacterial strains and their products was not examined.

Line 72. Delete the first comma.

Line 181. Replace “suggest” with “suggests”.

Line 254. Biding? Maybe “binding”.

Line 411. Delete “Patrick”.

Line 419. “C. elegans” in italics.

Line 467 (caption of fig. 2). Replace “celiac” with “coeliac”.

Line 473. Replace “is” with “are”.

Line 513. Replace “mice” with “mouse”.

Line 521. Full stop after “mice”.

Round 2

Reviewer 2 Report

Re: Manuscript ID: ijms-1728074

Only the following typing changes are suggested to improve the revised version of the manuscript.

Figure 1. Replace “microbiom” with “microbiome”. Replace “Proinflammator” with “Proinflammatory”. Replace “endothel” with “endothelium”. Replace “Submucosa” with “Mucosa” (the innermost layer is the tonaca mucosa; indeed, just under the epithelial cells there is the tonaca propria of the mucosa). Accordingly, in line 166 replace “submucosal” with “mucosal”. Of course, the same phenomena can spread into the submucosa, as well.

Into the capillary there are some microbes. Do you mean microbes or inflammatory mediators, as stated in the adjacent legend?

Line 122. Delete the second comma.

Line 140. Metabolom (?).

Line 177. Replace “separations” with “separation”.

Line 180. Replace “induce” with “induced”.

Line 396. Replace “this” with “these”.

Line 590. Delete the comma.

Line 592. Replace “increase” with “an increased”.

Line 594. Replace “have” with “has”.

Line 595. Replace “affects” with “affect”.

Line 603. Replace “faces” with “feces”.

Line 617. Replace “, represent” with “represents”.

Line 626. Replace “amount” with “amounts”.
